# Genomic Characterization and Expressional Profiles of Autophagy-Related Genes (*ATGs*) in Oilseed Crop Castor Bean (*Ricinus communis L.*)

**DOI:** 10.3390/ijms21020562

**Published:** 2020-01-15

**Authors:** Bing Han, Hui Xu, Yingting Feng, Wei Xu, Qinghua Cui, Aizhong Liu

**Affiliations:** 1Department of Economic Plants and Biotechnology, and Yunnan Key Laboratory for Wild Plant Resources, Kunming Institute of Botany, Chinese Academy of Sciences, Kunming 650204, China; hanbing@mail.kib.ac.cn (B.H.); xuwei@mail.kib.ac.cn (W.X.); 2University of the Chinese Academy of Sciences, Beijing 100049, China; 3College of Life Sciences, Yunnan University, Kunming 650091, China; xh15969478647@163.com (H.X.); fyt18087864922@163.com (Y.F.); 4Key Laboratory for Forest Resources Conservation and Utilization in Southwest Mountains of China, College of Forestry, Southwest Forestry University, Kunming 650201, China

**Keywords:** autophagy, castor bean, gene expression, seed development, seed germination

## Abstract

Cellular autophagy is a widely-occurring conserved process for turning over damaged organelles or recycling cytoplasmic contents in cells. Although autophagy-related genes (*ATGs*) have been broadly identified from many plants, little is known about the potential function of autophagy in mediating plant growth and development, particularly in recycling cytoplasmic contents during seed development and germination. Castor bean (*Ricinus communis*) is one of the most important inedible oilseed crops. Its mature seed has a persistent and large endosperm with a hard and lignified seed coat, and is considered a model system for studying seed biology. Here, a total of 34 *RcATG* genes were identified in the castor bean genome and their sequence structures were characterized. The expressional profiles of these *RcATGs* were examined using RNA-seq and real-time PCR in a variety of tissues. In particular, we found that most *RcATGs* were significantly up-regulated in the later stage of seed coat development, tightly associated with the lignification of cell wall tissues. During seed germination, the expression patterns of most *RcATGs* were associated with the decomposition of storage oils. Furthermore, we observed by electron microscopy that the lipid droplets were directly swallowed by the vacuoles, suggesting that autophagy directly participates in mediating the decomposition of lipid droplets via the microlipophagy pathway in germinating castor bean seeds. This study provides novel insights into understanding the potential function of autophagy in mediating seed development and germination.

## 1. Introduction

Cellular autophagy is a conserved process for turning over damaged organelles or recycling cytoplasmic contents in cells. It is considered an adaptive response to intracellular and extracellular environmental changes [1]. Although most autophagy-related genes (*ATGs*) were initially identified in yeast (*Saccharomyces cerevisiae*), studies have revealed that the autophagy process occurs widely in eukaryotic organisms, involving diverse and complicated physiological processes [2,3]. Generally, the autophagy process mobilizes two autophagic pathways: macroautophagy and microautophagy [4]. Macroautophagy is the principal pathway for decomposing substrates, using specialized double-membrane vesicles called autophagosomes for capturing materials [5], whereas microautophagy is a nonselective degradative process in which degradative materials are directly engulfed by lysosomes in mammals or vacuoles in plants [6]. Different *ATGs* carry out diverse functions at different stages of autophagy. Major components involved in autophagy include the ATG1/ATG13 kinase complex (that responds to nutritional cues provided by TOR, a conserved Ser/Thr kinase that regulates cell growth and metabolism), the ATG9/2/18 transmembrane complex (that supplies membranes to the autophagosome), the PI3 (phosphatidylinositol-3) kinase complex (that promotes vesicle nucleation), and the ATG8/ATG12 conjugation system (that assists with vesicle expansion and closure) [2]. In mammals, the autophagy process is functionally involved in regulating cell differentiation, cell senescence and apoptosis, further activating cellular response to starvation and adjusting the cell cycle [1].

In plants, studies have found that autophagy is broadly involved in diverse biological processes including metabolic nutrient recycling, leaf development and senescence [7]. Loss-of-function mutations of *AtATG8s* and *AtATG18s* have given rise to an induced expression of *AtSEN1* (a senescence marker gene) and led to the occurrence of leaf senescence and yellowing in Arabidopsis, suggesting that the autophagy process is critical in regulating leaf senescence and cell apoptosis in plants [8,9]. Many studies found that some *ATGs* (such as *ATG4*) were functionally required in mediating cell differentiation and elongation in Arabidopsis root tissues [10,11,12]. *ATG6* was functionally involved in regulating pollen vigor and germination [13,14]. There was additional evidence that plant cells cultured in vitro under nutrient starvation could produce autophagosomes to decompose cytoplasmic materials in vacuoles [15].

Autophagic-like pathways are also considered to be critical channels for transporting materials to the vacuole at different stages of plant development. Studies have demonstrated that autophagy is involved in regulating the accumulation of storage reservoirs (such as seed production) by altering nitrogen use efficiency in Arabidopsis and rice [16,17]. In tobacco [18] and maize [19], most *ATGs* were up-regulated in the late stage of developing seeds, implying that autophagy might have a role in the regulation of seed maturation. However, very little is known about the specific role of autophagy in this process. In addition, studies have found that autophagy can play a critical role in the decomposition of lipid droplets during the mobilization of fat (triacylglycerols) metabolism by sequestering a discrete region of the lipid droplets within animal cells. This selective autophagy specifically involved in lipid decomposition is called lipophagy [20,21]. Similar to the structure of lipid droplets, an oil body is the organelle for storing lipids within plant cells. There was evidence that showed lipophagy occurred in rice pollen maturation by decomposing oil bodies and providing energy for pollen development [22]. Usually, for oilseeds, the mobilization of storage oils mainly occurs in relation to seed germination in which storage oils are rapidly decomposed to release energy for activating seedling growth. However, whether the lipophagy process is required in the mobilization of storage oils in oilseeds remains unknown.

Although the genomes of many plants have been completely sequenced and ATG genes are easy to identify, ATG genes have only been identified in a few plants, such as *Arabidopsis thaliana* [23,24,25,26], rice [27], tobacco [18], maize [19], pepper [28] and grape [29]. Further investigations on the identification and characterization of *ATGs* at the genome level from diverse species are necessary to increase our understanding of *ATGs* in plants. Castor bean (*Ricinus communis* L., Euphorbiaceae) is one of the most important inedible oilseed crops and its seed oil is widely used in industry, especially for producing lubricating oil and biodiesel [30,31,32], because of its high content of the unique fatty acid ricinoleic acid. Although endosperm is unusual in dicots, castor bean is a typical member of this unusual group [33]. Thus, compared with *Arabidopsis thaliana*, castor bean is often considered a model system for studying seed biology of endospermous dicots [34,35,36]. In this study, based on the available genome data of castor beans, we identified all castor bean ATG genes, then characterized their structures and analyzed their expression over the castor bean life cycle. According to the expressional profiles of *ATGs* among different tissues, we found that autophagy might be required during seed development (in particular, seed coat development) and seed germination (the decomposition of storage oils in endosperm). This study provides a comprehensive profile of *ATGs* and their specific expression profiles among different tissues (in particular, for developing and germinating seeds) in castor beans, which adds to our understanding of the potential functions of autophagy during seed development and germination.

## 2. Results

### 2.1. Identification of Genes Encoding ATG Proteins in Castor Bean

Thirty-four putative *RcATGs* (Table 1) were identified by bioinformatic analyses of the published castor bean genomic sequences [37]. The 34 putative *RcATGs* vary from 115 aa to 1989 aa in length, from 13.2 kDa to 218.4 kDa in molecular weight and from 4.79 to 9.59 in protein isoelectric point (IP). To inspect the validation of the putative *RcATGs*, each RcATG amino acid sequence was checked in Pfam to test whether the conserved ATG domains were present in the RcATG sequence. Results showed that all *RcATGs* were validated. The 34 *RcATGs* were named and classified into 18 *ATG* families following the Arabidopsis category and nomenclature criteria [23,24,25,26] as listed in Table 1. Of them, *RcATG1* and *RcATG13* include three members (*RcATG1a*, *RcATG1b* and *RcATG1c*; *RcATG13a*, *RcATG13b* and *RcATG13c*), respectively; both *RcATG18* and *RcATG8* have seven members (*RcATG18 a-g*; *RcATG8 a-g*); and other *RcATGs* have only a single member. Based on the autophagy process pathway, the 34 *RcATGs* were predicted to participate in the different processes of autophagy. In castor beans, key catalytic enzymes that cover each necessary step in the autophagy pathways were identified, including induction of initiation for the ATG1/13 kinase complex (such as RcATG1 and RcATG13) and PI3 kinase complex (such as RcATG1, RcVPS15 and RcVPS34), membrane transport (such as the RcATG9, RcATG2 and RcATG18 complex) and elongation of autophagosome (such as the RcATG8 and RcATG12 conjugation system). These catalytic enzymes are essential for the occurrence of autophagy.

The sequence similarities of each homologous gene between castor beans and Arabidopsis were further analyzed, resulting in the high similarity of *ATGs* in sequence between castor beans and Arabidopsis. As shown in Figure 1, the gene structures of orthologous genes, including the distribution, number, length and splicing phase of intron/exon organization, are highly conserved between Arabidopsis and castor beans, although the intron length of some *RcATGs* varies, such as in *ATG2, ATG3*, *ATG4*, *ATG6*, *ATG9*, *ATG10*, *ATG11* and *ATG16L.* We also noted that the number of exons or introns for *ATG7, ATG12* and *VPS15* vary between species, which suggested that different alternative splicing existed in Arabidopsis and castor beans. For the *ATG1*, *ATG8*, *ATG13* and *ATG18* subfamilies, most members shared similar gene structure, despite slight variation in the number of exons or introns. These results suggest that the gene structures of *ATGs* were highly conserved during plant evolution.

To further understand the phylogenetic relationships of RcATGs, we conducted phylogenetic analyses in the multi-member subfamilies ATG1, ATG13, ATG18 and ATG8, using the amino acid sequences of RcATGs and their orthologs from Arabidopsis [23,24,25,26], rice [17], tobacco [18], maize [19] and yeast (*S. cerevisiae*) [3]. As shown in Figure 2, in the ATG1 subfamily, ATG1s were divided into two clades according to their evolutionary relationship (Figure 2A). The motif distribution in clade I is very similar compared to clade II. In the ATG13 subfamily, ATG13s were divided into three clades according to their sequence similarity (Figure 2B). Interestingly, RcATG13c was placed outside the clades because it did not contain many of the diagnostic motifs. In the ATG18 subfamily, ATG18s were clustered into four clades. Proteins in clade III and clade IV have more motifs compared with the other two clades (Figure 2C). In the ATG8 subfamily, the generated phylogenetic tree formed two distinct clades with well-supported bootstrap values (Figure 2D). It was clearly observed that almost all ATG8s were similar in length and had identical numbers of motifs, indicating that ATG8s were quite conserved across different species. However, 4 out of 6 ATG8s in clade II (66.7%) lack one or two motifs, whereas only ZmATG8e (4%) in clade I is absent. 

### 2.2. Expression Profiles of RcATGs in Various Tissues

Although the occurrence of the autophagy process requires the participation of multiple members within an activated autophagy network, a number of studies have shown that the expressional profiles of many ATG genes varied among different tissues [19], implying that different ATG genes might be involved in different biological processes. To understand the potential functions of *RcATGs*, we investigated the expression profiles of *RcATGs* in different tissues, including root, stem, leaf, seedling, ovule, capsule, inflorescence and seed across different development stages (seed1–seed5). The FPKM values of different tissues were acquired from the castor bean genome database (https://woodyoilplants.iflora.cn/). As shown in Figure 3, most *RcATGs* were differentially expressed among tissues. For instance, *RcATG10, RcATG18g* and *RcATG8a* were more highly expressed in root relative to other tissues; *RcATG1a, RcATG8c, RcATG101, RcATG18b*, *RcATG7, RcATG5* and *RcATG16L* were more highly expressed in reproductive tissues (ovule, capsule and inflorescence) and early seed (seed1 and seed2); *RcATG18d* and *RcATG13c* were more highly expressed in seed4 and seed2, respectively. During seed development, fewer genes were preferentially expressed in late seeds (seed3–seed5), especially fully mature seeds (seed5), while many *ATGs* were highly expressed in early seeds (seed1 and seed2). Depending on the gene expression pattern, hierarchical cluster analysis was performed to further explore similarities among samples. Twelve samples clustered into two major clades, designated as clade I and clade II. Clade II bifurcated into two subclades; clade IIa corresponded to vegetative tissues including root, leaf and stem; and clade IIb corresponded to early seed tissues (seed1 and seed2) and reproductive tissues, such as ovule, capsule and inflorescence. Clade I included all mature seed tissues, including seed3, seed4 and seed5. These results suggested that autophagy may not only vary greatly among different tissues but also participate in different biological processes at different seed development stages in castor beans.

### 2.3. Expression Profiles of ATGs in Developing Seeds

Unlike most dicotyledonous plants (such as Arabidopsis), the endosperm of the castor bean is persistent through seed development. After nuclear endosperm cells form cell walls and seed coats rapidly disappear, the endosperm cells divide quickly and eventually form a large nutritive tissue [36].

To investigate how autophagy was functionally involved in mediating seed development in castor beans, the expressional profiles of *RcATGs* were inspected by RT-qPCR in embryo, endosperm and seed coat tissues during seed development. As it is difficult to isolate endosperm and embryo tissues until about 20 or 25 DAP, we sampled the developing endosperm between 20 and 50 DAP, developing the embryo between 25 and 40 DAP. Because the seed coat will disappear after 30 DAP, we sampled the developing seed coat between 10 and 30 DAP. As shown in Figure 4, several *RcATGs* such as *RcATG1t*, *RcATG18d* and *RcATG18e* were highly expressed in developing endosperm tissues, while most *RcATGs* were not expressed or slightly expressed in developing embryo tissues. Interestingly, the expression levels of most *RcATGs* increased with seed coat development. Moreover, a large number of *RcATGs* were highly expressed in the late stage of seed coat development, in particular at the stage of 25–30 DAP, when the seed coat cellular tissue disappeared rapidly and was fiberized (see Appendix A). The disappearance of the seed coat, considered a programmed apoptosis [38], provides space for the formation of the castor bean endosperm.

### 2.4. Expression and Involvement of Autophagy in Mediating Storage Lipid Decomposition during Seed Germination

Increasing evidence shows that lipophagy is required for the decomposition of cellular lipid droplets [39,40]. However, little is known about whether lipophagy is involved in the decomposition of oil bodies during oilseed germination. To examine whether autophagy is involved in storage lipid decomposition during castor bean germination, we first inspected the change of oil content in seed germination. As shown in Figure 5, storage lipids were rapidly degraded starting 3 DAG. Upon examining the expression profiles of *RcATGs* by RT-qPCR in endosperms during castor bean seed germination, we found that most genes were not expressed (or marginally expressed) within the first two days and began to be up-regulated 3 DAG (Figure 6A), which is closely associated with the decomposition trend of storage lipids in seed germination (Figure 5). These results suggested that autophagy might have a role in mediating the decomposition of storage lipids in germinating castor bean seeds.

As mentioned above, lipophagy is involved in the breakdown of LDs for the maintenance of lipid homeostasis [40]. Usually, lipophagy occurs via two different pathways. One, termed microlipophagy, is where the LDs are directly degraded into fatty acids inside vacuoles. This has been observed in lower organisms such as single cell algae (*Auxenochlorella protothecoides*) and yeast (*S. cerevisiae*) [41,42]. The second is termed macrolipophagy, in which the degradation of LDs are mediated by autophagosomes (a double membrane organelle). This pathway has mainly been observed in the liver tissues of mammals and plant pollens [21,22]. To further inspect the potential lipophagy pathway occurring in castor bean germinating endosperm, we observed the morphological changes of LDs in endosperm throughout the seed germination under electron microscopy. As shown in Figure 6B, many LDs were directly bound with the vacuole surface and were eventually swallowed by vacuoles (Figure 6Ba–c) during seed germination. In particular, LDs that were swallowed by vacuoles were mainly observed after 4 DAG, which was associated with the expression of *RcATGs* in germinating endosperms. We did not, however, detect autophagosomes with a double-layer membrane structure. These observations indicated that the microlipophagy pathway mediated lipid degradation in germinating endosperm.

## 3. Discussion

There is increasing evidence that cell autophagy was extensively functionally involved in regulating plant development and responses to environmental stresses in a diverse set of plants [43,44]. As mentioned above, the castor bean has been considered a model material for studying seed biology in dicotyledonous plants [34,35,36]. Based on comprehensive analyses of identification and characterization of *ATGs*, we explored their putative functions in regulating seed development and germination in oilseed crop castor bean. To our knowledge, this is the first report on characterizing ATGs in the family Euphorbiaceae, which is an important group of plant resources. Here, we identified a total of 34 *RcATGs* that fall into four major cell autophagy processes, including the initial process-induced (ATG1/ATG13 kinase complex), membrane transport process (ATG9/2/18 complex), vesicle nucleation (PI3 kinase complex) and vesicle expansion. When inspecting the number of ATGs involved in the four conserved processes, we found 41 ATGs in maize [19], 32 in rice [17], 30 in tobacco [18], and only 12 in algae [41]. Variation in the number of ATGs might be related to the complexity and size of a plant’s genome. These observations indicated that although the autophagy pathway is highly conserved, processes and biological functions (such as some specific physiological processes within different tissues) probably vary in different plants. Studies have found that *ATG1s*, *ATG13s*, *ATG8s* and *ATG18s* often have multiple copies in plants [45]. These multiple copies comprise their own subfamilies. Here, gene structure and phylogenetic analyses revealed that although there are unique and specific motifs, the gene copies were phylogenetically orthologous within each subfamily. Moreover, for ATG subfamilies (ATG1s, ATG13s, ATG8s and ATG18s), some unexpected associations were found among different plants in Figure 2, such as placing ZmATG18b and RcATG18b in Clade I while placing AtATG18b and OsATG18b in Clade II, and placing ZmATG13a and OsATG13a in Clade I while placing AtATG13a and RcATG18a in Clade II, which suggests the phylogeny of ATG subfamilies is complex. 

Theoretically, the occurrence of the autophagy process requires the participation of multiple members within an activated autophagy network. A number of studies have shown that for a given member, its expressional profile varies among different tissues [19], implying that they might be involved in different biological processes. ATGs could be involved in regulating various processes of plant growth and development [43]. We identified differential expression of most ATGs in various tissues. We also noted that some ATGs were only expressed in specific castor bean tissues. The potential functions of these tissue-specific ATGs remain unknown in castor beans. In developing seeds, one of our main objectives was to explore the potential function of autophagy in regulating seed development. Our investigation found that many genes showed higher expression levels in the endosperm relative to the embryo. These *ATGs* that were highly expressed in the endosperm might be involved in regulating the metabolism and accumulation of storage materials [46]. Similarly, some *atg* mutants were previously found to cause a decrease in seed weight for Arabidopsis and maize [19,47]. These results mean that autophagy is involved in regulating storage material accumulation in seeds. In particular, our investigation found that most ATGs were highly expressed in the late stage of seed coat development, when the cell walls of seed coat cells experienced rapid lignification, resulting in hard and lignified seed coat tissues [36,48]. Previous studies had found that the programmed cell death caused by ricinosomes intensively occurred in the late stage of castor bean seed development (including the endosperm and seed coat tissues) [49,50]. Furthermore, the indicator gene *CysEP* for detecting the emergence of programmed cell death was highly up-regulated within the rapidly lignified castor bean seed coat tissues [50]. Thus, previous studies have confirmed that programmed cell death indeed occurs in seed coat tissues during the late stage of seed development in castor beans. However, the mechanism by which the process of programmed cell death was triggered remains unknown. Recent studies have revealed that autophagy is a critical factor in triggering programmed cell death [51,52]. Based on the intensive up-regulation of *RcATGs* in castor bean seed coat tissues, we infer that autophagy may be the trigger of programmed cell death during the late stage of castor bean seed coat development. It is further possible that autophagy is induced by programmed cell death for the clearance of terminally differentiated cells produced in the process of programmed cell death [53]. Many studies have found that *ATGs* were up-regulated at the late stage of seed development in different plants, such as tobacco [18] and maize [19]. The role of autophagy in regulating seed maturation might be common, despite diversity in the types of plant seeds involved. 

Another objective in our study was to explore the potential function of autophagy in regulating endosperm germination according to their expressional profiles in seed germination. The mature endosperm of castor bean seeds contains a large number of lipid droplets. These LDs are gradually decomposed during seed germination. For a long time, LDs were considered to be hydrolyzed in a lipolysis process by a series of lipases, such as ATGL (adipose triglyceride lipase), Tg13/Tg14 in yeast, HSL (hormone sensitive lipase) in mammals and GDSL lipases in plants [54,55]. After hydrolysis, the fatty acid product can provide energy to cells through β-oxidation in the mitochondria. Recent studies have shown that lipophagy is involved in the breakdown of LDs for the maintenance of lipid homeostasis [39]. One way in which lipophagy participates in degrading LDs is directly degrading them to fatty acids in the vacuoles, in lower organisms such as single cell algae (*Auxenochlorella protothecoides*) and yeast (*S. cerevisiae*), usually termed microlipophagy [41,42]. Another way in which lipophagy participates in degrading LDs is mediated by autophagosomes (a double membrane organelle). This is termed macrolipophagy, and has been observed by electron microscopy in plant pollen and in liver tissues of mammals [22,56]. Here, we observed degradation of LDs directly mediated by vacuoles and did not detect any double-membraned autophagosomes. The results of electron microscopy were similar to that of *Auxenochlorella protothecoides* and *S. cerevisiae* [41,42], suggesting that the role of autophagy in breaking down LDs might be largely mediated by the microlipophagy pathway in germinating castor bean endosperm. However, whether microlipophagy is a commonly used process in mediating the breakdown of LDs in other oilseeds remains unknown. Further research is required into microlipophagy in other oilseed species.

## 4. Materials and Methods

### 4.1. Plant Materials

The castor bean seeds var. ZB306 provided by Zibo Academy of Agricultural Science (Zibo, Shandong, China) were germinated and grown in the greenhouse under 13 h 28 °C day and 11 h 22 °C evening conditions. The root samples were collected at 14 DAG (days after germination) and the first internodes were collected as stems when grown to 5 cm in length. Leaves were collected two weeks after the blade appeared. Ovules were collected from the unfertilized fruits. Seed samples were collected from the greenhouse at different development stages from 10 to 50 days after pollination (DAP). Seed coats, endosperms and embryos were dissected after seeds were harvested, then immediately frozen in liquid nitrogen and stored at −80 °C.

### 4.2. Identification of ATGs in Castor Bean

Based on the castor bean genome (http://castorbean.jcvi.org/index.php), an extensive search was performed to identify all ATG genes in castor beans. The amino acid sequences of ATG genes generically are rather conserved with specific domains; therefore, the known ATG genes of Arabidopsis [23,24,25,26] could be used as queries for globally searching ATG genes in a given plant species. First, to identify the putative *ATGs* in castor beans, the BLASTP search was performed using the known *Arabidopsis* ATG protein sequences downloaded from the TAIR (https://www.arabidopsis.org/), as queries against the castor bean genome database. Genes with a significant E-value (<10^−5^) were collected while the redundant genes were discarded from our data set. After that, the online programs SMART (http://smart.embl-heidelberg.de/) and pfam (http://pfam.sanger.ac.uk/) were used to check the predicted ATG domains in these collected proteins. The genes which showed the most significant E-value with ATG domains were considered as putative *ATGs* of castor beans (*RcATGs*).

### 4.3. Bioinformatic Analysis and Phylogenetic Construction

The molecular weight (MW) and theoretical isoelectric point (PI) of different proteins were predicted using the Compute pI/Mw tool-ExPASy (http://web.expasy.org/compute_pi/). Intron/extron structure were analyzed using the online tool-GSDS 2.0 (http://gsds.cbi.pku.edu.cn/) based on genomic DNA sequences and the CDS sequences. Conserved motifs in RcATGs were identified by MEME online software (http://meme-suite.org/tools/meme) with the following parameters: normal motif discovery, optimum motif width (10–200 amino acids), 20 for maximum number of motifs and zero or one occurrence per sequence. For phylogenetic tree construction, amino acid sequences of ATGs from different species including Arabidopsis, rice, tobacco, corn, yeast were aligned using Clustal X version 2.1 first and then phylogenetic trees were constructed using MEGA X [57] by the neighbor-joining method with 1000 bootstrap replicates.

### 4.4. Analysis of RNA-seq Data

To investigate the expression profile of *RcATGs* in different tissues, the FPKM values of *RcATGs* in different tissues including root, stem, leaf, seedling, ovule, capsule, inflorescence and seed of different development stages (10 DAP for seed1; 20 DAP for seed2; 30 DAP for seed3; 40 DAP for seed4; 50 DAP for seed5) were acquired from the castor bean genome database (https://woodyoilplants.iflora.cn/). The pheatmap R package was used to map the heatmap and each *RcATG* was normalized to a Z-score based on FPKM values. Based on the ‘complete’ clustering method, hierarchical cluster analysis was performed.

### 4.5. RNA Extraction and qPCR

Total RNA was isolated from different tissues using RNAiso (TaKaRa). High quality RNA (three bands on agarose gel, absorbance ratio 260/230 1.8–2.0) were used to synthesize the cDNA with TransScript All-in-One First-Strand cDNA Synthesis SuperMix for qPCR (Trangene) following the procedure recommended by the manufacturer. The qPCR was carried out in a 20 µL PCR mixture containing 10 µL 2X TransStart Tip Green qPCR SuperMix (Transgene), 0.4 µL forward primer (10 µM), 0.4 µL reverse primer (10 µM), 1 ng cDNA prepared from different materials and variable ddH_2_O. The *RcACTIN2* was used as a control to normalize different samples. The specific PCR procedures are described as follows: denaturation at 94 °C for 30 s; then 45 cycles of 5 s denaturation at 94 °C and 30 s of annealing and synthesis at 60 °C. The primers for qPCR were designed using software primer premier 5 to ensure the specificity of amplified product, and are shown in Appendix A. All assays were performed at least three times from three biological replicates. The pheatmap R package was used to map the heatmap and each *RcATG* was normalized to a Z-score depending on the fold change.

### 4.6. Determination of Lipid Content

The method used for total lipid extraction was previously described by Xu et al. [58]. After grinding with liquid nitrogen for a few minutes, a fixed weight of dried sample (W) was added to 4 mL extracting solution (hexane/isopropanol = 3:2) (*v*/*v*). The mixture was fully shaken and then the supernatant was collected after centrifugation at 5000× *g* for 5 min. The precipitation was mixed with 4 mL extracting solution, and then the supernatant was collected again after centrifugation at 5000× *g* for 5 min. The combined supernatant was vacuum dried at a pressure of −60 kPa at 50 °C to a constant weight (w). The lipid content was equal to w/W × 100%. All assays were performed on at least three biological replicates.

### 4.7. Transmission Electron Microscopy

The castor bean seeds were cut into small pieces and fixed in a 0.05 M cacodylate buffer (pH 7.4) containing 2% glutaraldehyde and 4% paraformaldehyde for 12 h in an ice bath. The electron microscopy sections were prepared as described by Jiang et al. [59]. Cell images were taken using a transmission electron microscope (JEM-1200EX2, JEOL).

## Figures and Tables

**Figure 1 ijms-21-00562-f001:**
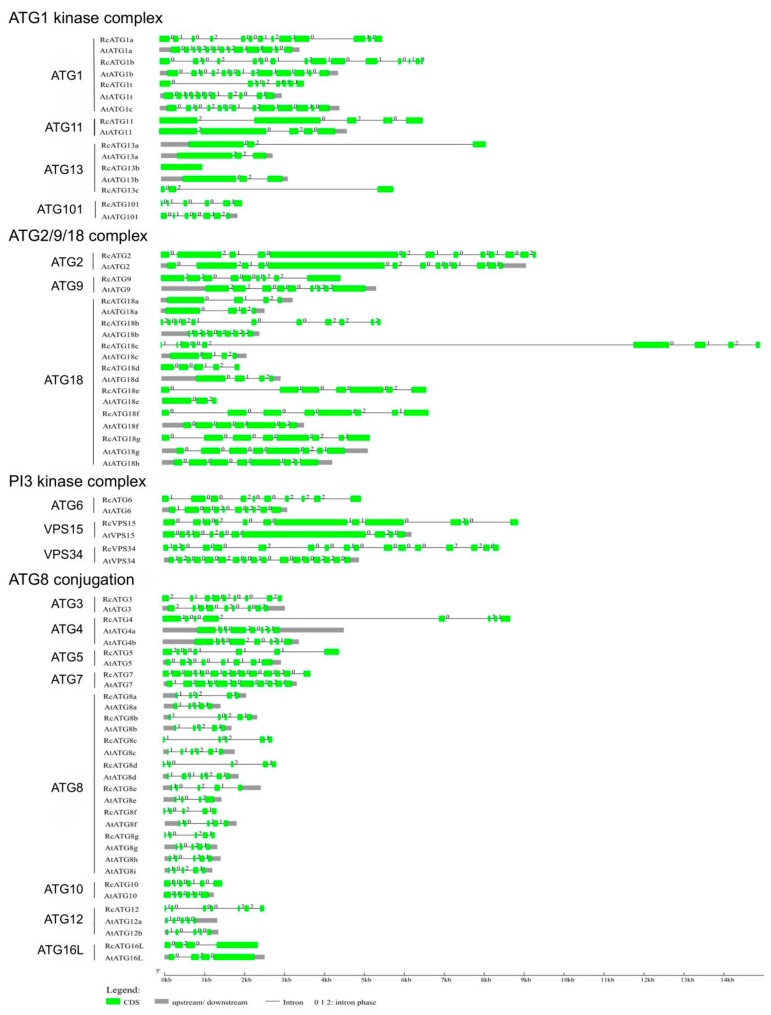
Exon/intron organizations of *AtATGs* and *RcATGs*. Green boxes indicate the exon regions and black lines indicate introns. The splicing phases: 0, splicing occurred after the third nucleotide of the codon; 1, splicing occurred after the first nucleotide of the codon; 2, splicing occurred after the second nucleotide of the codon.

**Figure 2 ijms-21-00562-f002:**
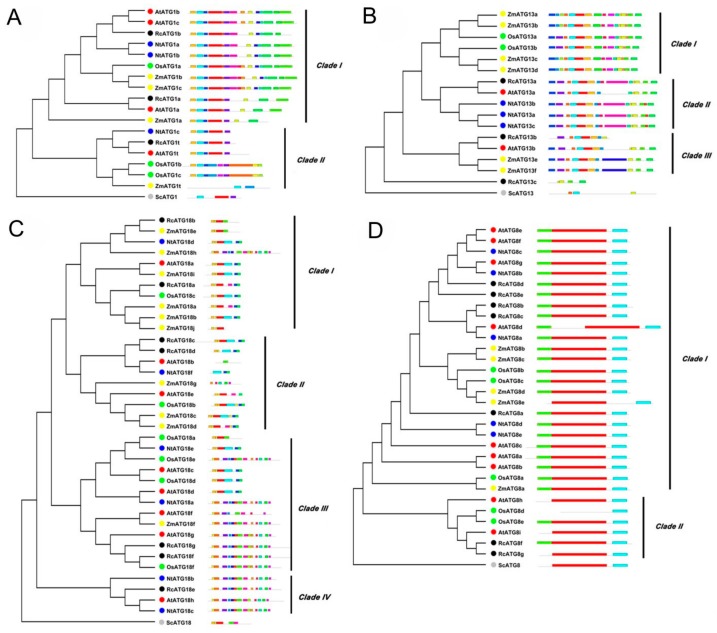
Bioinformatics analysis of autophagy subfamily genes in different species (i.e., Rc, *Ricinus communis L.*; At, *Arabidopsis thaliana*; Os, *Oryza sativa*; Nt, *Nicotiana tabacum*; Zm, *Zea mays*; Sc, *Saccharomyces cerevisiae*). (**A**) Phylogenetic tree and motif analysis of the ATG1 subfamily. (**B**) Phylogenetic tree and motif analysis of the ATG13 subfamily. (**C**) Phylogenetic tree and motif analysis of the ATG18 subfamily. (**D**) Phylogenetic tree and motif analysis of the ATG8 subfamily.

**Figure 3 ijms-21-00562-f003:**
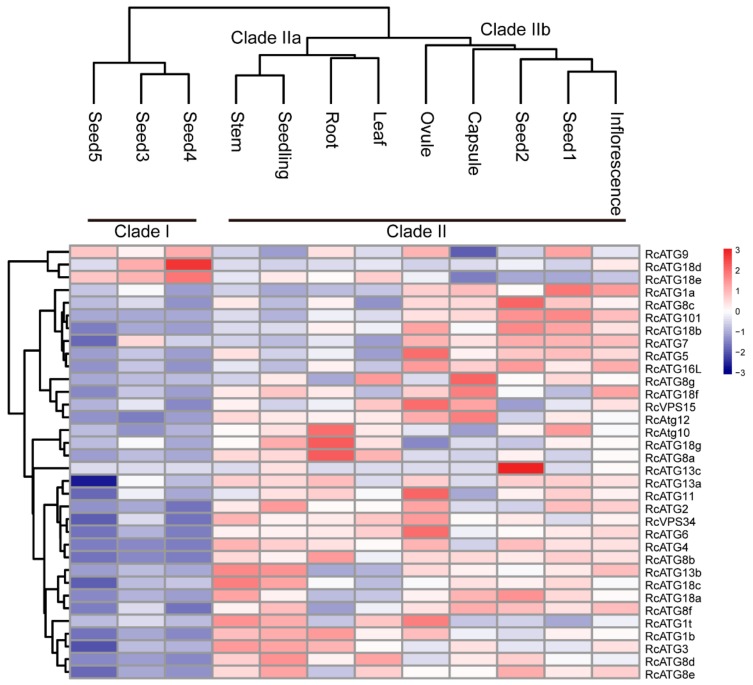
Expression profiles of *RcATGs* among different tissues. Blue box indicates the lower expression level of *RcATGs*, whereas the red box indicates the higher transcriptional level of *RcATGs*. The scale bar represents the relative expression level after normalization. Seed1–5 represent seeds at different development stages. The analysis was based on RNA-seq data downloaded from the castor bean genome database (https://woodyoilplants.iflora.cn/). Based on the ‘complete’ clustering method, hierarchical cluster analysis was performed.

**Figure 4 ijms-21-00562-f004:**
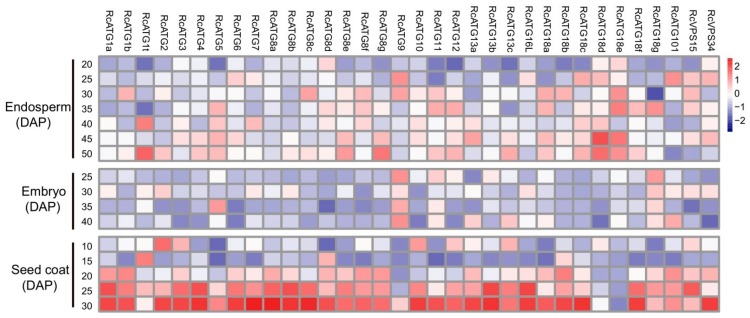
Expression profiles of *RcATGs* in castor bean seeds. The expression profile was constructed depending on the relative expression level of *RcATGs* in different materials. The expression is normalized to *RcACTIN2* and the data are means ± S.D. from three biological replicates. Blue boxes indicate the lower transcriptional level of *RcATGs*, whereas red boxes indicate the higher expression level of *RcATGs*. The scale bar represents relative expression level after normalization. DAP: days after pollination.

**Figure 5 ijms-21-00562-f005:**
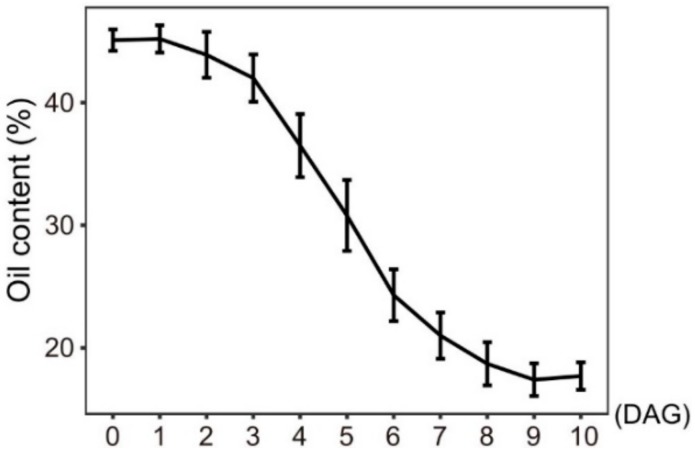
Oil content of germinating endosperm. DAG: days after germination. Bars = means ± SD from three biological replicates.

**Figure 6 ijms-21-00562-f006:**
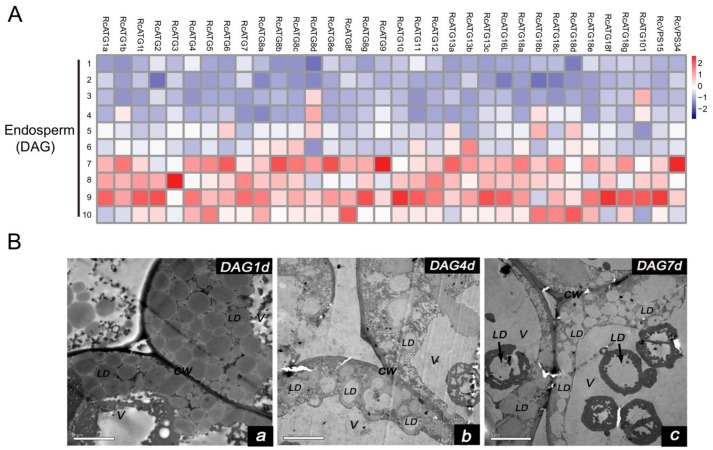
(**A**) Expression profiles of *RcATGs* in germinating endosperm. The expression profile was constructed depending on the relative expression level of *RcATGs* at different germination stages (DAG: days after germination). The expression is normalized to *RcACTIN2* and the data are means ± S.D. from three biological replicates. Blue boxes indicate the lower transcriptional level of *RcATGs* and red boxes indicate the higher expression level of *RcATGs*. The scale bar represents relative expression level after normalization. (**B**) Transmission electron microscopy of germinating endosperm (LD: lipid droplets; V: vacuole; CW: cell wall). a: DAG1d, b: DAG4d, c: DAG7d.

**Table 1 ijms-21-00562-t001:** The *ATGs* in castor beans.

Gene	Arabidopsis ID	Gene	Castor Bean ID	No. of Amino Acid Residues	Mw(kDa)	PI	E-value to Arabidopsis
**ATG1/13 Kinase Complex**							
AtATG1a	AT3G61960	RcATG1a	30076.m004465	676	75.1	5.90	4.4 × 10^−66^
AtATG1b	AT3G53930	RcATG1b	29973.m000397	694	76.9	6.44	7.6 × 10^−62^
AtATG1c	AT2G37840						
AtATG1t	AT1G49180	RcATG1t	29659.m000143	321	36.6	8.29	1.2 × 10^−88^
AtATG13a	AT3G49590	RcATG13a	30098.m001725	621	69.4	8.81	3.5 × 10^−103^
AtATG13b	AT3G18770	RcATG13b	30206.m000754	342	39.5	9.59	3 × 10^−83^
		RcATG13c	30206.m000753	220	23.9	5.42	2.1 × 10^−26^
AtATG11	AT4G30790	RcATG11	30114.m000524	1145	129.2	5.73	0
AtATG101	AT5G66930	RcATG101	29676.m001649	205	24.1	5.94	7.9 × 10^−76^
**PI3 Kinase Complex**							
AtVPS34	AT1G60490	RcVPS34	29631.m001016	813	93	6.48	0
AtATG6	AT3G61710	RcATG6	29742.m001430	523	59.9	5.85	2.1 × 10^−201^
AtVPS15	AT4G29380	RcVPS15	29912.m005424	1455	163.5	7.00	0
**ATG9/2/18 Complex**							
AtATG2	AT3G19190	RcATG2	29801.m003197	1989	218.4	5.43	0
AtATG9	AT2G31260	RcATG9	29584.m000241	864	99.7	6.32	8.4 × 10^−301^
AtATG18a	AT3G62770	RcATG18a	30129.m000356	447	49	6.59	3.6 × 10^−32^
AtATG18b	AT4G30510	RcATG18b	28612.m000123	349	38	8.86	7.1 × 10^−146^
AtATG18c	AT2G40810	RcATG18c	29729.m002396	598	67	7.56	1.2 × 10^−81^
AtATG18d	AT3G56440	RcATG18d	30128.m008811	330	36.6	9.33	1.6 × 10^−70^
AtATG18e	AT5G05150	RcATG18e	28166.m001060	891	96.9	6.44	0
AtATG18f	AT5G54730	RcATG18f	29703.m001515	1016	110.7	6.25	3 × 10^−115^
AtATG18g	AT1G03380	RcATG18g	30128.m008700	991	107.6	5.44	1 × 10^−113^
AtATG18h	AT1G54710						
**ATG8/12 Conjugation System**							
AtATG3	AT5G61500	RcATG3	30147.m014252	314	35.5	4.79	3.2 × 10^−150^
AtATG4a	AT2G44140	RcATG4	28966.m000547	489	53.9	5.13	2.3 × 10^−140^
AtATG4b	AT3G59950						
AtATG5	AT5G17290	RcATG5	30171.m000414	368	41.9	5.28	3.2 × 10^−115^
AtATG7	AT5G45900	RcATG7	29900.m001609	710	77.9	5.35	3.3 × 10^−235^
AtATG8a	AT4G21980	RcATG8a	29842.m003584	120	13.8	7.88	3.7 × 10^−53^
AtATG8b	AT4G04620	RcATG8b	30064.m000482	125	14.3	7.89	5.5 × 10^−52^
AtATG8c	AT1G62040	RcATG8c	30064.m000481	120	13.9	7.89	7 × 10^−52^
AtATG8d	AT2G05630	RcATG8d	29636.m000773	122	14.1	7.85	5.8 × 10^−48^
AtATG8e	AT2G45170	RcATG8e	30174.m008829	117	13.5	8.75	1.2 × 10^−45^
AtATG8f	AT4G16520	RcATG8f	30076.m004580	125	14.5	6.73	1.2 × 10^−33^
AtATG8g	AT3G60640	RcATG8g	29889.m003380	115	13.2	9.30	2 × 10^−31^
AtATG8h	AT3G06420						
AtATG8i	AT3G15580						
AtATG10	AT3G07525	RcAtg10	30178.m000879	233	26.9	5.25	7.3 × 10^−57^
AtATG12a	AT1G54210	RcAtg12	29706.m001289	154	17.4	8.58	1.3 × 10^−20^
AtATG12b	AT3G13970						
AtATG16L	AT5G50230	RcATG16L	30147.m013828	520	57.6	6.06	1.3 × 10^−203^

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
