# Peer review of "Genomic Characterization and Expressional Profiles of Autophagy-Related Genes (ATGs) in Oilseed Crop Castor Bean (Ricinus communis L.)"

_ijms, 2020, doi:10.3390/ijms21020562_

Round 1

Reviewer 1 Report

This manuscript is improved and questions have been satisfactorily answered. The authors have explained Seed 1, etc. from Figure 3 in the text, but I would still like to see some brief indication in Figure 3 or its caption as to what these designations are. As is, Figure 3 does not stand alone, which is recommended best practice.

Reviewer 2 Report

   Review of IJMS- 695650

Genomic characterization and expressional profiles of autophagy-related genes (ATGs) in oilseed crop castor bean (Ricinus communis L.).

Bing Han, Hui Xu, Yingting Feng, Wei Xu, Qinghua Cui, Aizhong Liu

The authors first used bioinformatics tools to identify all of the autophagy-related genes in the castor bean genome. They then characterized their intron-exon structures, and placed them in phylogenetic clades based on their sequence similarities to known autophagy-related genes from other species. They then used qRT-PCR to measure the expression of these RcATGs in various tissues at various stages of development including germinating endosperm. They then measured oil levels in germinating seeds and studied their ultra-structure by electron microscopy. They concluded that these genes helped the germinating seeds decompose storage lipids via microlipophagy in the vacuoles during seedling germination.

This study is interesting and provides potential insights into the role of autophagy in germination of oilseeds in general. There are still many mistakes in the English, but since my major criticisms have been addressed, I recommend publication after correcting the English and the issues listed below.

My most important remaining concern regards the criteria used to identify the RcATGs. As written in section 4.2 it seems that this was entirely based on BLASTP queries of the castor bean genome using all known Arabidopsis ATG protein sequences. However, they identified RcATGs with no Arabidopsis homologs, and also made some unexpected associations such as placing RcATG18b in Clade I while placing AtATG18b in Clade II very near RcATG18c and RcATG18d in figure 2. Similarly, RcATG13c did not seem closely related to any of the plant ATG13 genes, yet it was highly expressed at the seed 2 stage in figure 3 and in the embryo, endosperm and seed coat during the later stages of seed maturation according to figures 4 and 6A. Do they detect any similar sequences if they use this sequence as the query for a BLASTP search? Does this give some insight into its function? The authors need to explain their methods better in the “Methods” section and discuss these unexpected associations in their discussion.

It would also be useful to describe the phylogenetic relationship between Arabidopsis and Castor bean, since their paper relies on it so heavily. How long has it been since they diverged? Are there other plants with published genomic sequence data that are more closely related to Castor bean?

Another concern regards the conclusions from the electron microscopy. Many papers have published electron micrographs of germinating oilseeds from a wide variety of plants. How does the literature support the data presented in this report? Do previous studies support microlipophagy, or has anyone every reported double-membraned autophagosomes in germinating oilseeds from other plant species? These studies should be addressed in the discussion.

Many readers (including me) often look first at the figures and tables. Therefore, figure and table captions must provide enough information to understand and evaluate the figure or table without reading the narrative. Accordingly, the captions for figures 4 and 6A should explain that expression is normalized to RcACTIN2 and that data are means ± S.D. from 3 biological replicates.

Table 1 should be shown soon after it is described on line 101, rather than buried in the M&M.

There were many mistakes in the English. I list below some that caught my eye, but there are many more that should be corrected.

Line 14 should be: … autophagy is a widely-occurring conserved process for turning over damaged…

Lines 19 and 85: “non-edible” should be “inedible”

Line 20 should be: … coat, and is considered a model system for studying seed biology…

Lines 22-23 should be: The expression profiles of these RcATGs were examined using RNA-seq and real-time PCR in a variety of tissues.

Line 27 should be: Furthermore, we observed by electron microscopy that…

Line 35 should be: … autophagy is a conserved process for turning over damaged…

Lines 43-44 are confusing: what is an “oriental decomposition pathway? And what are “resolved organelles?”. Please rewrite for clarification!

Line 49: should “autophasome” be “autophagosome”?

Line 74 should be: … an oil body is the organelle for storing lipids within…

Lines 80-81 should be: Although genomes of many plants have been completely sequenced and ATG genes are easy to identify, ATG genes have only…

Line 87 should be: … of the unique fatty acid ricinoleic acid.

Lines 88-90 are confusing and should be rewritten to indicate that although endospermy is unusual in dicots, Castor bean is a typical member of this unusual group and is therefore a good model for studying the biology of this group of seeds.

Lines 91-94 are confusing and should be rewritten to indicate that the authors used the castor genomic sequence data available online to identify all castor bean ATG genes, then characterize their structures and analyze their expression over the castor bean life cycle. The authors need to also be careful to avoid confusing correlation with causation.

Line 98 should be “2. Results”

Lines 100-101 should be “Thirty four putative RcATGs (Table 1).were identified by bioinformatic analyses of the published castor bean genomic sequences [37].”

Line 104 should be: … test whether the conserved ATG domains were present in the RcATG sequence.…

Line 105-106 should be: The 34 RcATGs were named and classified into 18 ATG families following the Arabidopsis category and nomenclature criteria [23-26] as listed in Table 1.

Lines 107-116 are confusing and should be rewritten to indicate whether RcATG genes were identified encoding enzymes that could cover each necessary step in the macro or micro- lipophagic pathways. Can they rule out or rule in one of these pathways based on the presence or absence of key components?

Lines 117-126 are confusing and should be rewritten. Please emphasize what they have in common, and explain how the differences might have occurred.

Line 133 should be: … Interestingly, RcATG13c was placed outside the clades because it did not contain many of the diagnostic motifs…

Line 152 should be “2.2 Expression profiles of RcATGs in various tissues”

Lines 153-168 are confusing and should be rewritten for clarity and for correct grammar. The point is that although each ATG protein is part of a pathway, the expression patterns vary indicating that some may act in multiple pathways. Please avoid losing the reader, point out which genes that participate in the same process are co-regulated, and whether there are any genes that appear to participate in many processes or that show unexpected expression patterns for other reasons.

Line 169 should be “…clade IIa…”

Lines 176-179 should indicate that this analysis is based on RNAseq data downloaded from the

castor bean genome database (https://woodyoilplants.iflora.cn/) and that the data was analysed by the ‘complete’ clustering method.

Lines 181-191 are confusing and should be rewritten for clarity and for correct grammar.

Line 208 should be “Increasing evidence shows that lipophagy is required for the decomposition…”

Line 212 should be … As shown in Figure 5, storage lipids were rapidly degraded starting 3 DAG. Upon examining the expression…

Line 215 should be … be up-regulated 3 DAG …

Lines 223-224 should be .. Usually, lipophagy occurs via two different pathways. One, termed

microlipophagy, is where the LDs are directly degraded into fatty acids inside vacuoles..

Lines 226-230 are confusing and should be rewritten for clarity and for correct grammar.

Lines 245-256 are confusing and should be rewritten for clarity and for correct grammar.

Line 257 should be … size of a plant’s genome.

Lines 268-270 should be … We identified differential expression of most ATGs in various tissues. We also noted that some ATGs were only expressed in specific castor bean tissues. The potential functions of these tissue-specific ATGs remain unknown in castor bean.

Lines 272-273 should be … many genes showed higher expression levels in endosperm relative to embryo. These ATGs that were highly expressed in endosperm…

Line 277 should be … most ATGs were …

Lines 303-304 should be … One way in which lipophagy participates in degrading LDs is by directly degrading them to fatty acids in the vacuoles …

Lines 306-309 should be … Another way in which lipophagy participates in degrading LDs is mediated by autophagosomes (a double membrane organelle). This is termed macrolipophagy, and has been observed by electron microscopy in plant pollen and in liver tissues of mammals [22,56]. Here, we observed degradation of LDs directly mediated by vacuoles and did not detect any double-membraned autophagosomes.

Lines 317-318 should be … grown in the greenhouse under 13 h 28â—¦C day and 11 h 22â—¦C night conditions.

Lines 362-363 should be … follows: denaturation at 94℃ for 30s; then 45 cycles of

5s denaturation at 94℃ and 30s of annealing and synthesis at 60℃.

Line 381 should be … microscopy sections were…

Author Response

This manuscript is a resubmission of an earlier submission. The following is a list of the peer review reports and author responses from that submission.

Round 1

Reviewer 1 Report

   Review of IJMS- 654014

Genomic characterization and expressional profiles of autophagy-related genes (ATGs) in oilseed crop castor bean.

Bing Han, Hui Xu, Yingting Feng, Wei Xu, Qinghua Cui, Aizhong Liu

The authors first used bioinformatics tools to identify all of the autophagy-related genes in the castor bean genome. They then characterized their intron-exon structures, and placed them in phylogenetic clades based on their sequence similarities to known autophagy-related genes from other species. They then used qRT-PCR to measure the expression of these RcATGs in various tissues at various stages of development including germinating endosperm. They then measured oil levels in germinating seeds and studied their ultra-structure by electron microscopy. They concluded that these genes helped the germinating seeds decompose storage lipids via microlipophagy in the vacuoles during seedling germination.

This study is interesting and provides potential insights into the germination of oilseeds in general.

Unfortunately, it is not suitable for publication since it is not clear how many biological replicates and how many technical replicates were performed for each data point. Moreover, error bars are shown in figure 5, but the sample size and whether these represent S.D. or S.E. is not indicated. There must be at least 3 biological replicates for each sample, and 3 technical replicates of eahc qPCR reactions before it will be suitable for publication.

These omissions must be corrected before it can be considered for publication. In addition, the English must be extensively revised, since there are mistakes in nearly every sentence. I recommend revision by a native English speaker.

Reviewer 2 Report

The study "Genomic characterization and expressional profiles of autophagy-related genes (ATGs) in oilseed crop castor bean" performed by Han et al will be of interest to a broader section of the scientific community however the MS needs improvement. Below are my comments and concerns.

Title: Authors need to add the scientific name of castor bean in the title

Abstract: Cellular autophagy is a conserved and widely occurring decomposition process - better to write catalytic process instead of decomposition process

line 17 autophagy in mediating the plant growth and development, in particular, in recycling cytoplasmic content- should be plant growth and development, particularly in recycling ---

Line 21-22-Here, based on the castor bean genome 34 RcATGs genes were, in total, identified and their sequence structures were characterized- Reframe the sentence -should be "A total of 34 RcATG genes were identified in the castor bean genome ---

line 23-The expressional profiles of these RcATGs were inspected by RNA-seq and real-time PCR technique in dif- should be he expressional profiles of these RcATGs were (studied or examined) using RNA-seq and real-time PCR technique 

Introduction-The role of ATG genes need to be elaborated

Line No. 41. Reference No. 4 is wrong

Line No. 49 Reference is missing and please confirm the term ATG8/ATG12 conjugation system

Line No. 65 Reference no. 18 and 19 are wrong.

Line No. 80. Please mention originl reference of Arabidopsis thaliana (reference No. 23)

Need extensive English editing throughout the MS

Line No. 128 yeast (S. cerevisiae)-add reference

Line 155-158-To understand the potential functions of each RcATG involved in regulating castor bean growth and development, we investigated the expression profile of RcATGs in different tissues, including root, stem, leaf, seedling, ovule, capsule, inflorescence and seeds of different development stages(seed1-seed5) depend on the RNA-seq data downloaded from the castor bean genome database(https://woodyoilplants.iflora.cn/). Need to reframe the sentence

Line 163. Instead of seed 3 it must be seed 4 according to figure 3.

Line 165-166 Hierarchical cluster analysis was performed depend on the gene expression to further explore the similarity among samples. Correct the grammar or reframe the sentence

Line 170-171 These results suggested that autophagy may not only varies greatly among different tisues -Tissues need to correct spelling

Line 185-189- the seed coat will disappeared after 30 DAP, -the seed coat will disappear -Need English editing and split the sentence in two

Line 189-190-As shown in Figure 4, several RcATGs such as RcATG1t, RcATG18d, and RcATG18e were more highly expressed in developing endosperm tissues- were highly expressed -remove more

Line 191-194-were, as a whole, highly - remove the whole, correct the sentence and split in two

Line 211- started to be expressed or up-expressed- up-expressed or up-regulated

Line213-214 These results suggest that autophagy might be have a role in mediating the decomposition of storage lipids in germinating castor bean seeds

Line218-219 Usually, lipophagy in degrading LDs could be defined as
constituting two different pathways- reframe the sentence

Line 266-267 The potential mechanism of tissue-specifically expressed ATGs
remains unknown-reframe the sentence, remove hyphen from tissue-specifically

These more highly expressed- these highly expressed

Discussion needs improvement

Line 314- The root samples were collected at DAG (days after germination) 14- reframe the sentence -should be 14 DAG

line 315 when grown to 5cm in length give space between 5 and cm -5 cm

Line ing MEGA version 6.0 by - is it Mega X or mega 6.0 please add reference for the software

Neighbor-Joining method with bootstrap to be 1000 replicates- reframe the sentence.

Determination of lipid content- need to improve and elaborate it

How many sample replicates were used for qPCR

Authors can provide relative quantification graph of qPCR 

The authors need to elaborate on the method section of RNAseq- provide its accession number and how data was analysed.

How the normalization of replicates if used was performed.

What normalised values were used to generate heat map and which tool was used to generate heatmap.

Method section needs extensive improvement- 

Line no.189 and in Table1. Check whether its AtATG1t or AtATG1d

Reviewer 3 Report

My main concern with this manuscript is based on section 4.1 "Plant materials" where the authors state that most samples were taken from vegetative or pre-fertilization tissues, except for post-fertilization seed samples which were taken from field-grown plants. I don't think that the authors have addressed the weaknesses that this represents, e.g., how do the authors know that expression differences are due to stage/tissue and not environment (between greenhouse and field)? This needs to be addressed. More specific comments follow:

Line 46: What is TOR? Please define.

Line 87: The end of this sentence is unclear. Should “…typical endospermous oilseeds…” be “…typical of endospermous oilseeds…”?

Line 142: The splicing phase labelled “3” here should be “2”

Lines 145-146: Shouldn’t castor bean be added to the list of species in Figure 2?

Figure 3: Please define Seed1, Seed2, etc. in the caption. How many days after pollination for each?

Figure 3: Do the numbers on the scale bar refer to 3 times the expression of the actin gene? If so, please indicate in the caption. Or are the numbers arbitrary and just relative to each other? Please clarify.

Figure 3: Please briefly indicate, in the caption, what the horizontal and vertical trees indicate.

Line 174 (Figure 3 caption): Section 4.1 of the Materials and Methods indicates that most samples were taken from greenhouse-grown plants and seeds were collected from the field. So, please indicate this where “normal growth conditions” are stated, i.e., normal greenhouse and/or normal field conditions. Also see comments for lines 313-317.

Line 239: In the Figure 6 caption, please also define DAG as days after germination.

Lines 241-309 (Discussion section): Assuming that most tissues were harvested from greenhouse-grown plants, except for seeds which were harvested from field-grown plants: How can the authors compare expression in vegetative/pre-fertilization samples with post-fertilization samples when these samples are collected from such different environments? How do the authors know that differences in expression between vegetative/pre-fertilization and post-fertilization samples is due to stage rather than environment? Please address this concern.

Line 312: It’s unclear why “Castor” is used here instead of “castor bean” (also see line 362 and the “Castor ID” column heading in Table 1). If “Castor” is a cultivar/variety name, why is the variety also designated as “ZB306” in line 312? Please clarify.

Lines 313-317: This section says that most tissues were harvested from greenhouse-grown plants, except for developing seeds which were collected from the field. Is this accurate? If so, were the field-grown plants ZB306 as well? Also, I believe that castor bean has a mixed-mating system, so, were the field plants simply open-pollinated amongst each other? Or were they hand-pollinated by the researchers? If open pollinated, how could the researchers determine the exact day for each developing seed, so that days-after-pollination could be known for each sample?

Supplemental Table 1: Italicization of (F) and (R) is not consistent. Please revise.